# Foliar Lesions Induced by *Pestalotiopsis arengae* in Oil Palm (O × G) in the Colombian Southwest Palm Zone

**DOI:** 10.3390/jof10010024

**Published:** 2023-12-29

**Authors:** William Fabian Betancourt-Ortiz, Hector Camilo Medina-Cardenas, Jose Luis Padilla-Agudelo, Francia Helena Varon, Yuri Adriana Mestizo-Garzón, Anuar Morales-Rodríguez, Greicy Andrea Sarria-Villa

**Affiliations:** Pest and Disease Program—Phytopathology Area, Colombian Oil Palm Research Center (CENIPALMA), Centro Empresarial Pontevedra, Calle 98 N° 70-91, Piso 14, Bogotá 111211, Colombia; fabianb-1320@hotmail.com (W.F.B.-O.); hcmedinac@gmail.com (H.C.M.-C.); jlpadilla@cenipalma.org (J.L.P.-A.); franciavaron@gmail.com (F.H.V.); ymestizo@cenipalma.org (Y.A.M.-G.); amorales@cenipalma.org (A.M.-R.)

**Keywords:** leaf blight, molecular characterization, pathogenicity, phylogeny

## Abstract

In Colombia, plantings with the oil palm hybrid between *Elaeis oleifera* × *Elaeis guineensis*, known as O × G hybrid, have increased due to its tolerance to bud rot. Despite this, different degrees of foliar necrosis, chlorosis, and leaf blight have been reported in some cultivars; therefore, this work aimed to diagnose this problem. We visited plantation plots with palms exhibiting the mentioned symptoms and collected 21 samples of affected tissues in different disease states. The affected tissues were examined and seeded in a culture medium. Pathogenicity tests were performed and the isolates were characterized by culture and morphological and molecular features. *Curvularia*, *Colletotrichum*, *Phoma*, and 25 *Pestalotiopsis*-like fungi were isolated from the foliar lesions. In the pathogenicity tests, the symptoms observed in the field were reproduced with MFTU01-1, MFTU12, and MFTU21 isolates, which were identified at the species level through a sequence analysis of three genes (*ITS*, *TUB2*, and *TEF1-α*) as *Pestalotiopsis arengae* with an identical level of 99% based on the results of BLAST and phylogenetic tree analyses. The remaining 22 *Pestalotiopsis*-like non-pathogenic isolates were identified as species of *Neopestalotiopsis* and *Pseudopestalotiopsis*. The direct association of *P. arengae* with the disease was confirmed via molecular detection in affected tissues in 15 of 21 samples collected for this evaluation. This is the first report of *P. arengae* as the causal agent of foliar lesions in O × G hybrid oil palm in Colombia.

## 1. Introduction

Among oil crops, oil palm registers the highest yields per hectare [1]. Colombia is currently the fourth largest palm oil producer in the world after Indonesia, Malaysia, and Thailand [2]. The palm industry in Colombia generates approximately 177,400 direct and indirect jobs and contributes to 6% of the country’s GDP [3]. Oil palm is affected by biotic and abiotic agents. Bud rot (BR) disease caused by the oomycete *Phytophthora palmivora* is one of Colombia’s most limiting phytosanitary problems [4]. In the southwestern Colombian palm-growing region (Tumaco, Nariño), this disease reached epidemic proportions in the first decade of the 2000s and devastated the 36,000 hectares planted with *Elaeis guineensis* cultivars [5]. As a result of the BR epidemic in the Colombian Southwest Palm Zone, renewals have been made with hybrid cultivars of *Elaeis oleifera* × *Elaeis guineensis* (O × G) since 2008. The genetic solution seems promising: incorporating the resistance of the American oil palm (*E. oleifera*) into the African oil palm (*E. guineensis*) generates the O × G interspecific hybrid that is tolerant to diseases such as BR [6,7,8,9,10]. 

Some plantations in the zone have reported emerging problems in the different hybrids, such as the drying of lower leaves, chlorosis of the canopy, and leaf blight, with the latter having the highest incidence. These symptoms have been associated in *E. guineensis* with the disease known as *Pestalotiopsis* or leaf blight, which involves a complex of fungi, including *Pestalotiopsis palmarum* and *P. glandicola*, and other genera like *Helminthosporium* sp., *Curvularia* sp., *Colletotrichum* sp., *Phyllosticta* sp., and *Macrophoma palmarum* [11,12,13]. Traditionally, the development of the disease has been linked to primary damage caused by insects with sucking, chewing, and scraping habits [13,14,15,16,17,18]. However, there are few studies where isolations and pathogenicity tests have been performed to verify the direct association of microorganisms with the development of foliar lesions. 

The taxonomic classification at the species level within the *Pestalotiopsis* genus has traditionally been made according to the host from which fungi were isolated, the conidia morphology, the color intensity of the conidia median cells, and molecular analyses based on rDNA sequences [19,20]. However, recent studies by Maharachchikumbura et al. [21] and Ayoubi and Soleimani [22], supported by the molecular analysis of multiple genes (which included the internal transcribed spacer of ribosomal DNA (*ITS*) [23], beta-tubulin (*TUB2*) [24], and translation elongation factor 1-alpha (*TEF1-α*) [25]) have contributed to improving the traditional classification system of the species, indicating that the trait of median cell coloration is not a reliable taxonomic trait in the discrimination of species. 

Several reports have indicated *Pestalotiopsis* species as plant pathogens in tropical and subtropical countries [26,27]. In Colombia, Solarte [28] studied the genetic diversity of the genera *Pestalotiopsis* and *Neopestalotiopsis* related to the Scab disease of Guava (*Psidium guajava* L.). In Italy, Ismail [29] reported that *P. uvicola* and *P. clavispora* caused gray leaf spots on mangoes (*Mangifera indica* L.). Keith [30] reported that *P. clavispora*, *P. microspora*, and *P. disseminata* caused guava scab (*Psidium guajava* L.) in Hawaii. Maharachchikumbura et al. [31] reported that *P. semarangensis* caused fruit rot of Pomarrosa (*Syzygium samarangense*) in Thailand. Maharachchikumbura [32] reported that *Neopestalotiopsis*, *Pestalotiopsis,* and *Truncatella* species are associated with trunk disease in grapevine (*Vitis vinifera* L.) in France. Additionally, Maharachchikumbura et al. [33] associated *Pseudopestalotiopsis ignota* and *Ps. camelliae* with grey blight disease of tea (*Camellia sinensis*) in China. 

In oil palm crops in Colombia, some studies by Jimenéz O. [34] have associated *Pestalotiopsis* species with leaf blight. For example, on the cultivar *E. guineensis*, a complex of fungi was identified in association with leaf blight or añublo foliar. However, there are no studies on the interspecific hybrid O × G. In China, Shen et al. [35] reported *Pestalotiopsis microspora* causing foliar lesions in African oil palm; in Thailand, *P. theae* has been reported to cause foliar lesions in *E. guineensis* [36]; and in Queensland, Australia, Fröhlich [37] associated *P. phoenicis*, *P. elastica*, *P. theae*, *P. adjusta*, and *P. palmarum* with the foliar lesions presented in the coconut tree (*Cocos nucifera*). In Colombia, leaf blight of oil palm has been registered as a disease of economic importance. It generally affects palms older than two years [34]. Jiménez and Reyes [12] compared the production of slightly affected palms (19–24% defoliation) with severely affected palms (55–66% defoliation) and reported losses in bunch production close to 36% per year, equivalent to 5.2 tons of bunches/hectare per year and the depressive effects persisted until 33 months after having carried out control of the disease. The problems at the leaf level observed in hybrid cultivars in the Colombian Southwest Palm Zone and the scarce information on the association of the microorganisms involved in this pathology have motivated this research, with the purposes of (i) identifying and describing the symptomatology of this foliar disease; (ii) performing pathogenicity tests with the obtained microorganisms with similar characteristics to *Pestalotiopsis*; and (iii) characterizing the causal agent. These are all indispensable steps for the management practices that will be necessary to mitigate the incidence and severity of the disease. 

## 2. Materials and Methods

### 2.1. Description of Symptoms and Foliar Tissue Sampling

This research was carried out in plantations in the Colombian Southwest Palm Zone located in the Department of Nariño, specifically in the Municipality of Tumaco (1°48′0″ N, 78°45′0″ W); according to the Meteorological Station located in the El Mira Research Center of Agrosavia (observation window of the last 20 years), Tumaco has an average annual precipitation of 2910 mm, distributed mainly in a single wet season between the months of January to June. Likewise, the average annual temperature is 26.6 °C with a maximum of 27.8 °C and a minimum of 25.6 °C (Figure 1). 

Under field conditions, initial observations were made to select hybrid palms to be evaluated, identifying the leaflet lesions, describing the initial symptoms, and marking the progress toward the most affected leaves. Additionally, photographic records of the symptoms were taken at different stages of the development of the disease. Affected tissues were collected at different stages of infection from the oil palms sampled at the study site, which were planted between 2007 and 2010. The collected samples were transported under refrigeration to Cenipalma’s plant pathology laboratory for immediate processing. 

### 2.2. Isolation and Purification of Microorganisms

To identify the fungal microorganisms present in the foliar tissue, imprints of the field-collected material were created. To induce sporulation, the tissue was placed in a moist chamber consisting of 28 oz aluminum trays. Each tray contained a paper towel moistened with sterile, distilled water and was covered with a piece of tulle to prevent direct contact between the leaves and the water. The trays were then placed inside Ziploc© bags. Finally, part of the tissue was processed following the traditional method for fungal isolation, which consists of washing the sample with tap water, drying it, and selecting affected tissue from the area of disease progression in 0.3 to 0.5 cm long pieces. Subsequently, the samples were disinfected with 70% alcohol for 30 s and 1% sodium hypochlorite for 1 min and rinsed three times with sterile distilled water. The samples were then dried and seeded in several culture media: potato dextrose agar (PDA), vegetable juice (V8) agar, water agar, corn meal agar, malt extract agar, and oatmeal agar [38]. The samples were incubated at 24 °C (±2 °C) for 24 h under white light and 24 h in darkness. When the reproductive structures of the fungi were formed, monosporic cultures were obtained by directly transferring the conidia to Petri plates with PDA medium [39]. 

### 2.3. Culture and Morphological Characterization

The culture and morphological characterization for *Pestalotiopsis*-like isolates were performed using the methods used to identify fungal genera of agricultural importance [38,40,41]. Regarding the culture characterization, the development of the microorganism in the PDA culture media was described according to the colony’s color, shape, margin elevation or surface edge, texture, and conidiomata production [30]. 

All the isolates were identified using the morphological basis of the reproductive structures: the length and width of the conidia, lengths of apical and basal appendages, the number of basal and apical appendages, the numbers of cells and septa present in the conidia, and finally the color exhibited by the median cells [21,42]. 

The conidia were taken from the conidial suspension and observed under an Olympus BX40 light microscope with 40× magnification. Photographic records were taken using an Olympus DP73 camera (Olympus Corporation, Tokyo, Japan). Biometric variables were estimated with the CellSens^®^ software cellSens V1.14 (https://www.olympus-lifescience.com.cn/en/sofware/cellsens/, accessed on 10 July 2021). The 95% confidence interval was found for each variable, and the maximum and minimum values and the mean values of 30 measurements were estimated. The measurements of the structures were compared using an analysis of variance (ANOVA) and the means were separated according to Tukey’s test (*p* < 0.05) using the statistical software SAS 9.0.

### 2.4. Molecular Identification and Phylogenetic Analysis

The biomass development of 25 *Pestalotiopsis*-like isolates was produced in potato dextrose broth (Potato Dextrose Broth, DIFCO) with shaking at 110 rpm for seven days. From fresh fungal mycelia, genomic DNA was extracted using the QIAGEN DNeasy Plant Mini kit (Hilden, Germany), following the manufacturer’s instructions. The final concentration of the DNA was 100–200 ng/μL. The amplified gene regions included the internal transcribed spacer of ribosomal DNA (*ITS*) using primers ITS4 and ITS5 [23], beta-tubulin (*TUB2*) with primers Bt2a and Bt2b [24], and translation elongation factor 1-alpha (*TEF1-α*) with primers EF1-526F and EF1-1567R [25]. The reactions were incubated in a T3 thermocycler (Biometra, Göttingen, Germany). For each reaction, a total volume of 25 µL was used containing 12.5 µL of the kit’s GoTaq^®^ Green Master Mix (Promega, Madison, WI, USA), 9.5 µL of nuclease-free water, 0.3 µM of each primer, and 30 ng of genomic DNA. The reaction conditions included an initial denaturation at 95 °C (5 min), followed by 35 cycles of denaturation at 94 °C for 1 min, binding at 52, 55, and 54 °C for *ITS*, *TUB2*, and *TEF1-α*, respectively, for 30 s, and extension at 72 °C for 1 min with a final extension at 72 °C for 10 min. The PCR products were run on a 1% agarose gel. After electrophoresis, the gels were visualized under UV light. The bands at the expected sizes were cut directly from the gel, purified using the QIAquick Gel Extraction kit (Qiagen, Hilden, Germany), and sent to Universidad Nacional de Colombia for sequencing with the primers used in the PCR analysis. 

The sequences obtained with each primer were edited using the BioEdit 7.0.5.3 program [43], and the consensus sequence was constructed with the CAP contig assembly accessory for each isolate. Once the sequences were obtained, their identity was confirmed by comparing them with the GenBank database based on *ITS*, *TUB2*, and *TEF1-α*. The sequences generated in this study were supplemented with additional sequences obtained from other related strains of *Pestalotiopsis*, *Neopestalotiopsis*, and *Pseudopestalotiopsis* species from GenBank based on the literature reported for phylogenetic analyses (Appendix A). Subsequently, sequence alignment was carried out using the MUSCLE algorithm included in the MEGAX program [44]. The statistical method was the neighbor-joining method of the consensus sequences of the *ITS* gene, and evolutionary distances were computed using the Kimura 2-parameter method. This analysis involved 103 nucleotide sequences. There were a total of 464 positions in the final dataset. The support of the nodes was determined using the bootstrap method with 1000 repetitions. Our consensus sequences are available under the accession numbers for the *ITS* region, *TUB2*, and *TEF1-α* in Section 3.5. Also, we developed a phylogenetic tree for each genus (Appendix A).

### 2.5. Pathogenicity Tests

For the pathogenicity tests, we used 7-month-old O × G hybrid cultivar Sinú (cerete) × Deli Yangambí. The tests included microorganisms isolated from symptomatic tissue isolates such as *Pestalotiopsis*-like sp. (25 isolates), *Curvularia* sp. (1 isolate), *Colletotrichum* sp. (1 isolate), and *Phoma* sp. (1 isolate). Preliminary tests were conducted by using different inoculation methods to select the most effective one in reproducing the symptoms [35,45,46]. A wound was necessary to develop lesions, acting as a facilitator for the entry of the microorganism. 

The inoculation method used a 5 mm mycelial agar disc from the edge of a colony of the different isolates from an approximately 10-day-old culture. A sterile entomological pin (0.40 mm diameter by 37 mm long) was used to create the wound in the leaf tissue before placing the plug. The same procedure was used with pathogen-free PDA discs for the control treatments. The plants were kept under moist chamber conditions for 48 h in both cases. Subsequently, they were maintained in semi-controlled conditions under a shade house (70% dark) at an average temperature of 28–30 °C combined with a misting system to promote infection. The development of the infection over time was analyzed using ImageJ^®^ program developed by Wayne Rasband (National Institutes of Health, Bethesda, MD, USA, https://www.nih.gov/, accessed on 10 July 2021). Ten repetitions were performed for every isolate, and one palm with two lanceolate and inoculated leaves was a repetition. The size of the lesion (mm^2^) was measured at 3, 7, and 31 days after inoculation (dai) and analyzed utilizing a repeated measures analysis (RM-ANOVA). The averages were analyzed using Tukey’s test (*p* < 0.05) using the statistical software SAS 9.0.

### 2.6. Primer Design and Molecular Detection of Pestalotiopsis Arengae in Foliar Tissues

To confirm the presence and direct association of *P. arengae* with foliar lesions from the field, the specific primers used were based on the comparison of the *ITS* region of *P. arengae* isolates and different sequences of *Pestalotiopsis*, *Neopestalotiopsis*, and *Pseudopestalotiopsis* deposited in GenBank and obtained in our study. The primers Parenga155ITSF (5′-TGTAGCGCCCTACCCTGGAACGA-3′) and Parenga534ITSR (5′-GGCTAAGGACGCTGCAACTCCAGT-3′) were designed with the AmplifX tool (https://inp.univ-amu.fr/en/amplifx, accessed on 10 June 2021) to amplify 380 bp PCR products. The specificity of the primers was evaluated at an in silico level, and the homology of the amplicon of the expected size was compared against other species reported in the NCBI databases using the BLASTn tool (BLAST+ 2.12.0, https://blast.ncbi.nlm.nih.gov/Blast.cgi, accessed 12 September 2021). The reaction conditions included an initial denaturation at 95 °C (5 min), 40 cycles of denaturation at 94 °C for 30 s, binding at 56 °C for 30 s, and extension at 72 °C for 1 min, and a final extension at 72 °C for 10 min. For this evaluation, at least 1 sample from each of the 16 plantations visited was used, and 21 samples were processed for the molecular detection of the pathogen. The methodology for the extraction of total genetic material was the same as that carried out for the fungal isolates and followed the manufacturer’s instructions. The PCR products obtained were sequenced, and their identity was confirmed by comparison with the GenBank database.

## 3. Results

### 3.1. Description of Symptoms in Naturally Infected Farms 

Lesions were frequently distributed on both sides of the leaf sheet (top and underside), with oval or elongated shapes, an aqueous and translucent appearance, and surrounded by an undefined chlorotic halo. As the lesions progressed, they turned irregular and reddish brown, delimited by a dark brown halo and accompanied by a chlorotic halo. In advanced stages of the infection, lesions coalesced and formed concentric zones with brown coloration and shades of grey; these lesions always exhibited a yellowish orange halo. When the lesions coalesced, the leaflets became chlorotic, and finally, the leaves presented large necrotic areas with greyish drying and loss of foliar tissue (Figure 2). 

Forty-seven samples of diseased plant material from 16 plantations in the three regions (subzones (Figure 1)) that comprise the oil palm area of southwestern Colombia were collected. These regions contain different *E. oleifera* × *E. guineensis* (O × G) genotypes under different crop management and edaphoclimatic conditions. 

### 3.2. Isolation of Microorganisms

A total of 25 *Pestalotiopsis*-like isolates were obtained from symptomatic plants from different oil palm hybrid cultivars. In addition, other fungi such as *Curvularia* sp. (one isolate), *Colletotrichum* sp. (one isolate), and *Phoma* sp. (one isolate) were isolated from the symptomatic plants. Some of these microorganisms have been associated with foliar diseases in oil palms. The information on the *Pestalotiopsis* isolates is available in Table 1. 

### 3.3. Cultural Characterization

The 25 isolates with ten days of growth in PDA were distributed in six morphotypes. Morphotype 1 had 10 isolates (40%) with a cotton texture, irregular edges, high and limited appearance, white-grey (above) and white-salmon (reverse) colony, and development of surface conidiomata. Morphotype 2 had five isolates (20%) with a cotton texture, irregular edges, high and limited appearance, white-cream (above) and white-salmon (reverse) colony, and development of surface conidiomata. Morphotype 3 had four isolates (16%) with a cotton texture, irregular edges, fat, and extended appearance, white-cream (above) and white-cream–red-orange (reverse) colony, and development of surface conidiomata. Morphotype 4 had three isolates (12%) with a cotton texture, irregular edges, high and limited appearance, white-grey (above) and white-grey (reverse) colony, and development of surface conidiomata. Morphotype 5 had two isolates (8%) with a cotton texture, irregular edges, fat, and extended appearance, white-cream (above) and white to salmon (reverse) colony, and development of surface conidiomata. Morphotype 6 had one isolate (4%) with a cotton texture, irregular edges, high and limited appearance, white-grey (above) and brown-reddish (reverse) colony, and development of surface conidiomata (Table 2).

### 3.4. Morphological Characterization

Based on the initial morphological characterization, the isolates belonged to two genera: *Neopestalotiopsis* (19 isolates) and *Pestalotiopsis* (six isolates); this classification was assigned based on the number of cells, the number of septa, and the color exhibited by the median cells of the conidia. Additionally, the length and width of the conidia, apical and basal appendages length, median cell length, and the number of basal and apical appendages were determined and are available in Table 2. Furthermore, a molecular analysis (see Section 3.5) enabled the identification of a third genus, known as *Pseudopestalotiopsis* (comprising three isolates that were initially classified under the genus *Pestalotiopsis* based solely on their morphological characteristics).

There were significant differences between genera for each biometric variable evaluated (*p* < 0.05). The measurements of the conidia are shown in Table 2. The isolates belonging to the genus *Neopestalotiopsis* showed fixed, solitary to slightly gregarious conidiomata, which were semi-immersive and brown to black in color. The conidial exudate was dark brown and globose. The conidia had a fusiform, ellipsoid shape and were straight to slightly curved, formed by five cells, and divided into four septa. Measurements of the length and width of the conidia ranged from (17.7) 23.7 to 24.3 (48.5) $\bar{x}$ ± SD = 24.0 ± 4.0 μm and (4.5) 6.3 to 6.4 (10.2) $\bar{x}$ ± SD = 6.4 ± 0.8 μm. The basal and apical cells were hyaline. The length of basal and apical appendage measurements ranged from (2.5) 4.9 to 5.0 (9.2) $\bar{x}$ ± SD = 4.9 ± 1.2 μm and (3.0) 16.9 to 17.6 (34.1) $\bar{x}$ ± SD = 17.2 ± 4.5 μm, respectively. The median cells were dark brown, clear brown, and olive-brown colored; they had 1–4 unbranched apical appendages and a single basal, tubular, unbranched appendage. The length of medium cell measurements ranged from (11.3) 15.6 to 16.1 (23.7) $\bar{x}$ ± SD = 15.8 ± 2.6 μm.

The isolates belonging to the genus *Pestalotiopsis* presented globose, clavate, solitary, or aggregated conidiomata, and a semi-immersed, dark brown to black mucilaginous appearance. The conidia presented an ellipsoid shape and were straight to slightly curved and slightly contracted at septa, with values that ranged between (23.2) 27.8 to 28.7 (33.1) $\bar{x}$ ± SD = 28.3 ± 2.1 μm length and (5.8) 6.6 to 6.8 (7.9) $\bar{x}$ ± SD = 6.7 ± 0.5 μm width. The conidia were made up of five cells and were divided into four septa: the apical and basal cells were hyaline with a cylindrical shape, and the median cells were concolorous brown, with septa darker than the rest of the cells ranging from (15.5) 17.9 to 18.5 (21.5) $\bar{x}$ ± SD = 18.2 ± 1.3 μm length; 2–3 unbranched filiform apical appendages ranging from (9.0) 14.7 to 15.7 (19.8) $\bar{x}$ ± SD = 15.2 ± 2.4 μm length; and one basal, simple, tubular, unbranched apical appendage that was (2.7) 4.4–4.9 (8.4) $\bar{x}$ ± SD = 4.6 ± 1.2 μm in length (Figure 3). 

The conidiomata of *Pseudopestalotiopsis* isolates were immersed in a viscous, globose, and glossy mass. Morphologically, the conidia were fusiform, ellipsoid, straight to slightly curved, with four septa that were (22.7) 27.7–29.0 (40.5) $\bar{x}$ ± SD = 28.3 ± 3.2 μm in length × (4.1) 6.5–6.9 (8.8) $\bar{x}$ ± SD = 6.7 ± 1.0 μm in width; apical and basal hyaline cells; median cells that were concolorous pale brown and (16.7) 18.9–19.5 (23.8) $\bar{x}$ ± SD = 19.2 ± 1.5 μm in length with 2–4 tubular apical appendages (mainly three) that were unbranched and (14.8) 21.1–22.7 (31.5) $\bar{x}$ ± SD = 21.9 ± 3.9 μm in length; and a single basal appendage that was tubular, unbranched, centered, and (3.2) 4.8–5.4 (8.4) $\bar{x}$ ± SD = 5.1 ± 1.3 μm in length. 

### 3.5. Molecular Characterization

The PCR for the 25 isolates produced amplicons for rDNA between 530 and 630 bp in size for *ITS*, 440–470 bp for *TUB2*, and 850–1030 bp for *TEF1*. Based on the initial comparison of our sequences against GenBank using BLAST, it was revealed that all isolates in this study belonged to three genera: there were three isolates of *Pestalotiopsis* (MFTU01-1, MFTU12, and MFTU21), three of *Pseudopestalotiopsis* (MFTU14-1, MFT014-2, and MFTU34-2), and 19 isolates of *Neopestalotiopsis* (MFTU04-3, MFTU06-1, MFTU06-2, MFTU06-3, MFTU07-1, MFTU07-2, MFTU18, MFTU25, MFTU34-3, MFTU35, MFTU36, MFTU37, MFTU38, MFTU39-1, MFTU39-2, MFTU40-1, MFTU41, MFTU43, and MFTU44-2) (Table 3). To establish the relationship between species within each genus, we utilized a concatenated dataset comprising 27 taxa for *Pestalotiopsis*, 8 taxa for *Pseudopestalotiopsis*, and 41 taxa for *Neopestalotiopsis*. The type strain of *Seiridium phylicae* (KC005788) served as the outgroup taxa (Figure 4).

The phylogenetic analysis allowed the grouping of three isolates corresponding to *Pestalotiopsis* (MFTU01-1, MFTU12, and MFTU21), with a relative support on the branches of 99% with the holotype strain of *Pestalotiopsis arengae* (CBS 331.92) that was reported by Maharachchikumbura (Appendix A) [47]. For the *Pseudopestalotiopsis* group, the isolate MFTU34-2 formed one clade with *Pseudopestalotiopsis ampullacea* (NKT0P03) that was reported by Liu [48]. The isolates denominated MFTU14-1 and MFTU14-2 were grouped with *Pseudopestalotiopsis cocos* (CBS 272.29) that was obtained from *Cocos nucifera* in Indonesia and reported by Maharachchikumbura [47] (2014) (Appendix A). 

In the case of the 19 remaining isolates, they were distributed into three main clades with reliability support on the branch above 85% with different species of the genus *Neopestalotiopsis*. The total isolates of this group were clustered into three clades defined in this study as *Neopestalotiopsis* sp. I, *Neopestalotiopsis* sp. II, and *Neopestalotiopsis* sp. III. The isolates MFTU07-1, MFTU38, and MFTU41 were grouped with reference sequences like *Neopestalotiopsis cubana* (CBS 600.96), *Neopestalotiopsis protearum* (CBS 114178), and *Neopestalotiopsis saprophytica* (CBS 115452), which were previously published by Maharachchikumbura [47] and Monclova-Santana [49], respectively. In the phylogenetic tree for each genus, we found that MFTU07-1 formed a clade with the strain *Neopestalotiopsis* sp. 10 (CBS 110.20) [47]. The isolates MFTU06-3 clustered in a clade (II) with *Neopestalotiopsis surinamensis* (CBS 450.74). The isolates denominated MFTU04-3 and MFTU06-1 were grouped in the sub-clade (I) recently published by Solarte et al. (2018) [28] with *Neopestalotiopsis* sp. 14 (strain VR1ep) obtained from guava fruit in the Department of Valle del Cauca (Colombia) (Appendix A). Also, MFTU41 was grouped with two published strains, VTman6 and SVsnp3, based on sequences from the same study. 

### 3.6. Pathogenicity Tests

To confirm pathogenicity, seedlings of the cultivar Sinú (cerete) × Deli Yangambí were inoculated with a mycelial PDA disc on wounded tissue. *Cuvularia* sp. caused small circular lesions that were initially chlorotic and then turned brown but always retained their circular development. The leaves inoculated with *Colletotrichum* sp. presented elongated brown necrotic lesions with irregular growth. Finally, *Phoma* sp. presented small yellowish-brown circular lesions with a reddish border [50]. The symptoms produced by these three pathogens were different from the symptoms observed in the leaf spot in this work (Appendix A). 

The 25 *Pestalotiopsis* isolates showed different degrees of aggressiveness, measured as lesion size (mm^2^). Three days after inoculation (dai), the first lesions were oval and brown, surrounded by a chlorotic halo like the symptoms in the field. The development of lesions beyond the inoculation site was observed only for three isolates (MFTU01-1, MFTU12, and MFTU21) (Figure 5), and the characteristics of the lesions were like those reported for the genus *Pestalotiopsis* by Suwannarach [36]. Only one isolate (MFTU04-3) induced small lesions but did not show development over time, reaching only 2% of the affected leaf surface. In addition, the cultivars Brazil × 7 pollen africans, Coarí × La Mé, and Manaos × Compacta were evaluated (Appendix A).

The isolates MFTU01-1, MFTU12, and MFTU21 initially induced circular or oval brown lesions surrounded by an irregular chlorotic halo. The lesions developed until 31 dai when they turned dark brown at the edge with grey tones at the center; in more advanced stages, the lesions presented a necrotic area in the center, as well as tissue loss and the development of concentric black-dot-like conidiomata; the chlorotic irregular halo was present in all stages of the diseases (Figure 5). There were significant differences in the size of the lesions (*p* < 0.05) compared with the other isolates. The average lesion sizes at the end of the test were 365, 346, and 336 mm^2^ for MFTU12, MFTU01-1, and MFTU21 isolates, respectively (Figure 6). Control plants did not show any development lesions. Re-isolation of the microorganism was obtained from the lesions, fulfilling Koch’s postulates.

### 3.7. Molecular Detection of Pestalotiopsis arengae in Foliar Tissues

The specific primers designed in this study detected the presence of *P. arengae* in naturally infected leaf tissues from the field via PCR. The expected specific size of the PCR products was visible on an agarose gel for 15 of the 21 newly collected and evaluated samples (Figure 7). The comparative analysis of the sequenced samples against the GenBank database confirmed with an identity of 100% that *P. arengae* (E-value: 0.0) is the causal agent of leaf blight in the oil palm hybrid O × G in Colombia. The sequences of the amplicons obtained with the specific primers were deposited in GenBank under accession numbers MZ045516–MZ045530.

## 4. Discussion

This study revealed two important aspects of the disease known as leaf blight in the cultivation of oil palms in Colombia. The first one was the pathogenic association of the genus *Pestalotiopsis* with the development of foliar lesions in the interspecific O × G hybrid of oil palm. Secondly, not all fungal microorganisms obtained from foliar lesions such as blight are etiologically related to the direct development of the disease. 

In Colombia, there is little information about the presence, prevalence, and etiology of the main foliar diseases that affect O × G hybrid cultivars. In this study, the genotypes planted in Tumaco, Colombian Southwest Palm Zone, had lesions that caused drying and necrosis of leaves, mainly in the lower leaves of the palms. In *E. guineensis,* several microorganisms have been associated with leaf spots, with a greater prevalence of *Pestalotiopsis*, which is considered the causal agent of leaf blight. In this sense, Jiménez and Reyes [12] conducted a study in the zone of Puerto Wilches (Santander, Colombia) and found that some *Pestalotiopsis* species in association with the insect *Leptopharsa gibbicarina* (Hemiptera: Tingidae) [15] can induce necrosis, drying, and defoliation because the insect generates the wounds necessary for the penetration and colonization of the pathogen. On the other hand, Martínez and Plata [51] reported the *Pestalotiopsis* genus as the cause of leaf blight. They associated it with the damage caused by several lepidopteran insects commonly found on *E. guineensis* in some plantations of Colombia. Although these authors pointed out the role of lepidopteran insects in the development of the disease, isolation and pathogenicity tests of *Pestalotiopsis* were not performed; thus, a direct association of the microorganism with the disease cannot be defined.

In our study, several fungal organisms were observed and isolated from the samples taken from the different hybrid cultivars in Tumaco, including *Curvularia*, *Colletotrichum*, *Phoma*, and *Pestalotiopsis*-like fungi. In Colombia, some references indicate the association of these fungal microorganisms with leaf spots in *E. guineensis* [12,52,53], but there is no information on their pathogenicity or aggressiveness in hybrid O × G cultivars. This association described in *E. guineensis* with *Curvularia*, *Colletotrichum*, and *Phoma* was based on observing initial lesions that did not continue a progressive and infective process over time (Jiménez and Reyes, 1977). In the pathogenicity test carried out in this research, placing the agar plugs with mycelial growth and conidiomata on the wound made in the tissue was an effective method for reproducing symptoms. This finding coincides with that reported by Elliott [54], Keith [30], and Solarte [28], indicating that *Pestalotiopsis* requires a wound to penetrate the plant, which is a necessary factor for the development and severity of the disease; for this reason, the microorganism is considered weak because it requires a facilitator in its infectious process [13,14,34,55].

Different *Pestalotiopsis* species have been reported to affect palms of *E. guineensis* in different countries. Shen [35] reported for the first time that *P. microspora* causes leaf spot diseases in China. Likewise, Suwannarach [36] reported *P. theae* as the causal agent of this foliar disease in Thailand. In this study, 3 isolates, denominated MFTU01-1, MFTU12, and MFTU21, out of 25 samples, were identified within the genus *Pestalotiopsis.* These isolates exhibited morphological characteristics consistent with *Pestalotiopsis arengae* and were further confirmed through molecular tools [47]. For the species delimitation, a comparison of the *ITS* sequence of our isolates with *P. arengae* (CBS 331.92) revealed that 590/593 base pairs are similar with only two gaps between them and a query cover up to 98% [56]. Our isolates initiated the infective process and developed lesions in the inoculated plants. The species of *P. arengae* in the O × G hybrid oil palm constitutes a new record for this host and fungi species as they have yet to be reported earlier on this crop. Due to the low number of pure isolates, it was necessary to implement other methodologies, such as PCR, to detect the presence or absence of this microorganism. The low isolation rate is attributed to the presence of other secondary and opportunistic microorganisms in the affected tissues. These microorganisms were able to grow in the culture media used in our study because it was not selective for the isolation of *P. arengae*. The direct association of the presence of *P. arengae* in affected leaf tissue from the field was confirmed via molecular detection using PCR with species-specific primers designed in this study. The species registered as the causal agent of leaf blight in Colombia has been previously reported in Singapore as *Arenga undulatifolia* by Maharachchikumbura [47], and it is closely related to *P. anacardiacearum* isolated from mango in China [46] and *P. hawaiiensis* isolated from *Leucospermum* sp. in Hawaii [46]. 

The differentiation of *Pestalotiopsis* species with morphological characteristics is complicated due to the overlapping ranges of values. However, our pathogenic morphotypes obtained from the O × G hybrid oil palm present similar characteristics to those obtained and reported in another palm species, *Arenga undulatifolia* [47]. In the PDA culture medium, the colonies of MFTU01-1, MFTU12, and MFTU21 (Figure 3) exhibited cream and pale white tones on the top and the bottom of the plate; the margins were wavy with slightly aerial mycelium and gregarious conidiomata; and the conidia had a fusiform, ellipsoid, straight to slightly curved shape and were formed by five cells and four septa. The apical and basal hyaline cells were cylindrical, and the median cells were concolorous. The conidia had 2–3 unbranched filiform apical appendages and one simple, tubular, unbranched basal appendage. In contrast, differences in conidia width and the apical and basal appendage length were observed. Maharachchikumbura [47] reported that *P. arengae* conidia were 7–9.5 (10) μm wide, $\bar{x}$ ± SD = 8 ± 0.4 μm, while the reproductive structures reported in this work were slightly thinner at (5.8) 6.6–6.8 (7.9) $\bar{x}$ ± SD = 6.7 ± 0.5 μm. The apical appendages observed in the isolates MFTU01-1, MFTU12, and MFTU21 were (9.0) 14.7–15.7 (19.8) $\bar{x}$ ± SD = 15.2 ± 2.4 μm in length, which are longer than those reported in *A. undulatifolia* ((4) 4.5–11 (12) μm long, $\bar{x}$ ± SD = 7.3 ± 1.3 μm). The length of the basal appendage ((2.7) 4.4–4.9 (8.4) $\bar{x}$ ± SD = 4.6± 1.2 μm) is longer than that reported for the morphotype distributed in Singapore (1.5–3 μm). The conidia length ((23.2) 27.8–28.7 (33.1) $\bar{x}$ ± SD = 28.3 ± 2.1 μm) and long media cells ((15.5) 17.9–18.5 (21.5) $\bar{x}$ ± SD = 18.2 ± 1.3 μm) were similar to those recorded by Maharachchikumbura [47] for *P. arengae* ((24) 25–32 (33) $\bar{x}$ ± SD = 27.6 ± 2 μm for the conidia length and (17) 17.5–21.5 (22) μm long, $\bar{x}$ ± SD = 19 ± 1.3 μm, for the median cells). 

Through the combined sequence analysis of *ITS*, *TUB2*, and *TEF1-α* genes, it was possible to resolve the identity of a few species of the *Neopestalotiopsis* genus obtained in this study; however, some isolates were not grouped with any previously reported sequences. These isolates formed clades denominated in our study as *Neopestalotiopsis* sp. I and III and could be defined as new species; similar results were obtained in Colombia by Solarte [28] and in Mexico by Gerardo-Lugo [57]. For the other isolates grouped in the genus *Pseudopestalotiopsis*, it was possible to resolve their taxonomic identification by combining the analyses of the three genes and considering their phylogenetic grouping with two previously reported species. 

This is the first report of *Pestalotiopsis arengae* causing foliar lesions in the O × G hybrid in Colombia or elsewhere. In fact, this species has not yet been associated with damage or leaf spots in other crops of agricultural importance in Colombia. In this study, we also described the use of PCR to detect and confirm the presence of *P. arengae* from affected foliar tissues. Likewise, we showed the diversity of species of the genus *Neopestalotiopsis* and *Pseudopestalotiopsis* without any detrimental effect on the leaves of the O × G hybrid in Colombia. In future studies, we expect to analyze the association of the *Neopestalotiopsis* and *Pseudopestalotiopsis* isolates with the development of lesions in joint inoculations with the isolates of *Pestalotiopsis* to continue studying their biology, epidemiology, and the economic impact on the oil palm crops.

## Figures and Tables

**Figure 1 jof-10-00024-f001:**
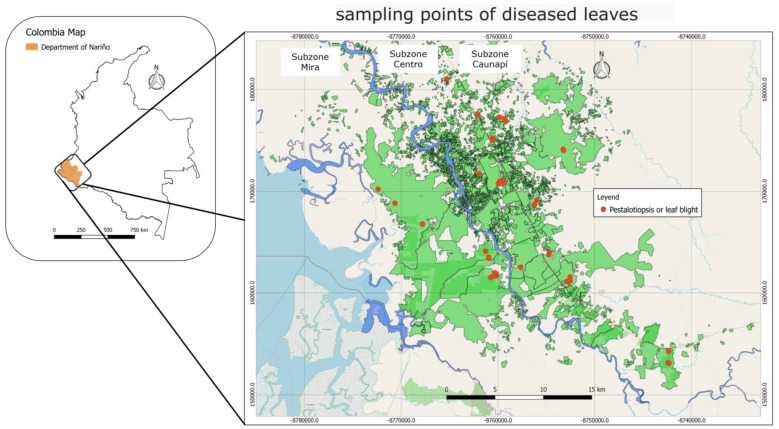
Location of oil palm plantations containing the interspecific hybrid O × G sampled for Pestalotiopsis or leaf blight.

**Figure 2 jof-10-00024-f002:**
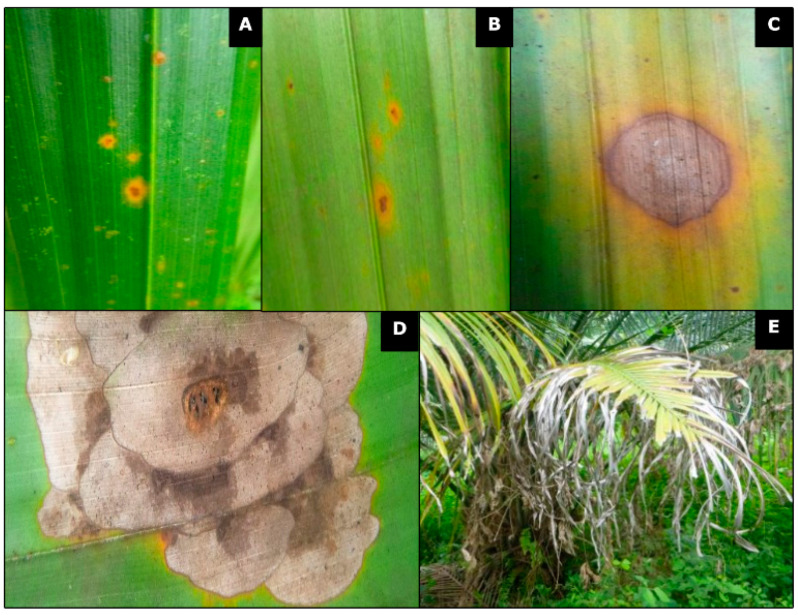
Development of foliar lesions in hybrid oil palm. (**A**) (top) and (**B**) (underside): Oval or elongated brown lesions surrounded by irregular chlorotic halo. (**C**): Progressive lesion delimited by a dark brown halo and accompanied by a chlorotic halo. (**D**): Coalesced irregular lesions forming concentric zones. (**E**): Chlorosis of leaflets followed by drying.

**Figure 3 jof-10-00024-f003:**
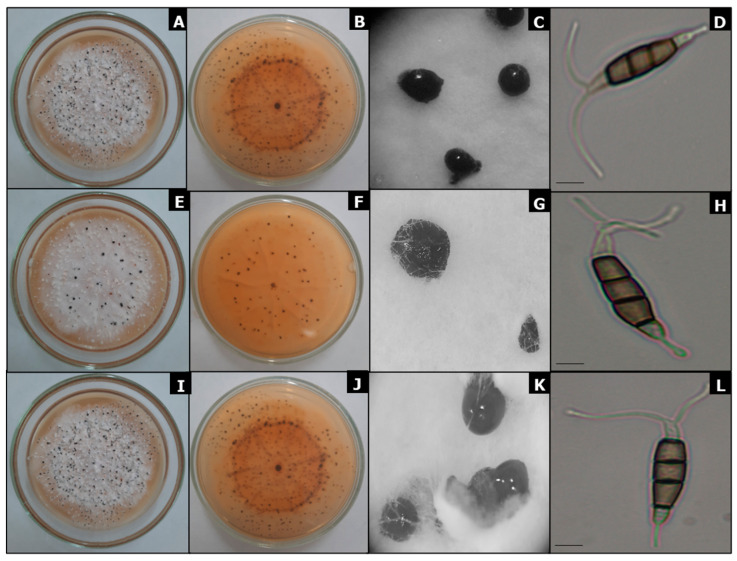
*Pestalotiopsis arengae.* From left to right, sporulation in Petri dishes containing potato dextrose agar (front and back, respectively), conidiomata in PDA, and conidia. Scale bars = 5 µm. Isolates: MFTU01-1 (**A**–**D**); MFTU12 (**E**–**H**); and MFTU21 (**I**–**L**). (**A**–**J**): View of colony growth in Petri dish; upper- and underside. (**C**–**K**): Conidiomata. (**D**–**L**): Conidia.

**Figure 4 jof-10-00024-f004:**
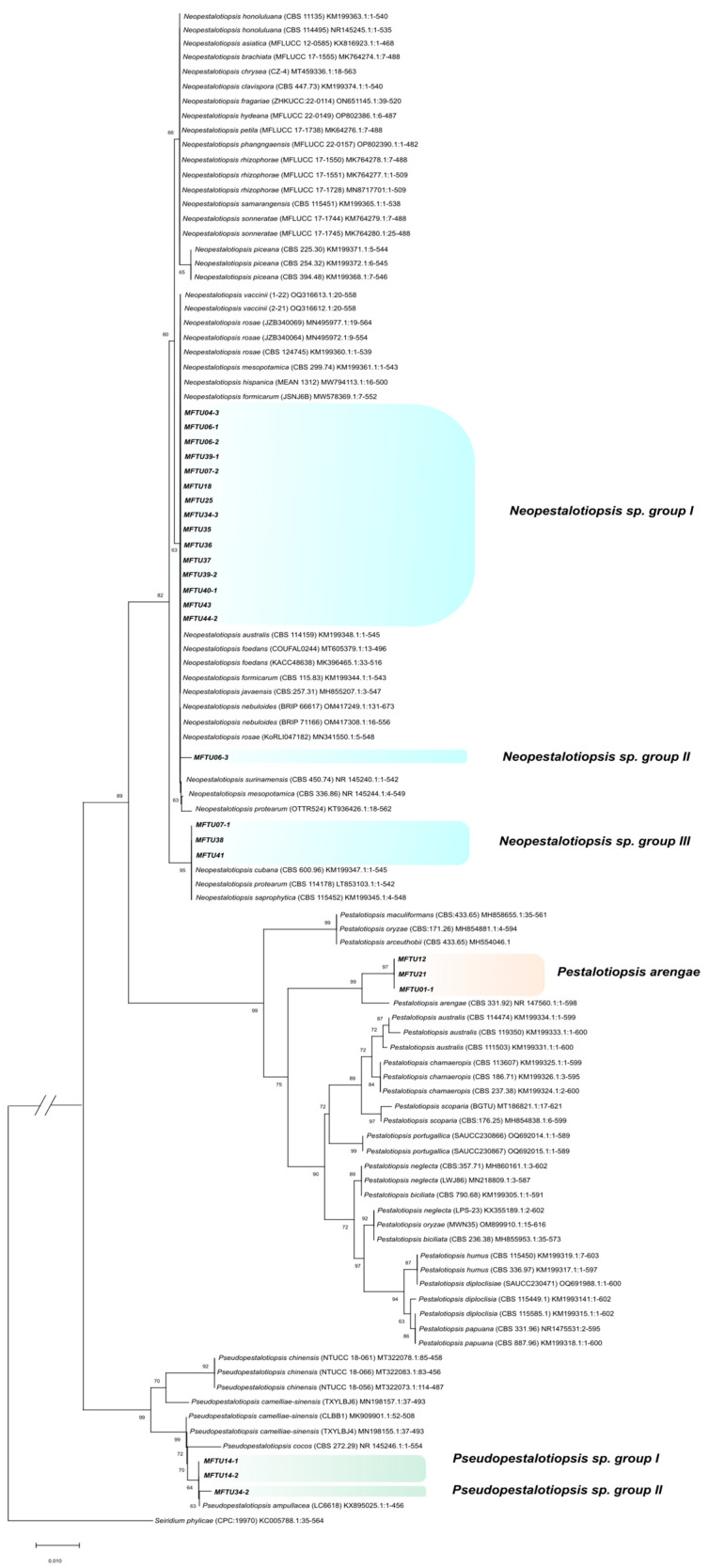
Phylogenetic tree obtained using the neighbor-joining method of the consensus of the *ITS* region of some species of the genera *Pestalotiopsis*, *Pseudopestalotiopsis*, and *Neopestalotiopsis*. The evolutionary distances were computed using the Kimura 2-parameter method. This analysis involved 103 nucleotide sequences. There were a total of 464 positions in the final dataset. Evolutionary analyses were conducted in MEGA X. The isolates corresponding to this work are indicated in bold. The species Seiridium phylicae (CPC:19970) was included as an outgroup.

**Figure 5 jof-10-00024-f005:**
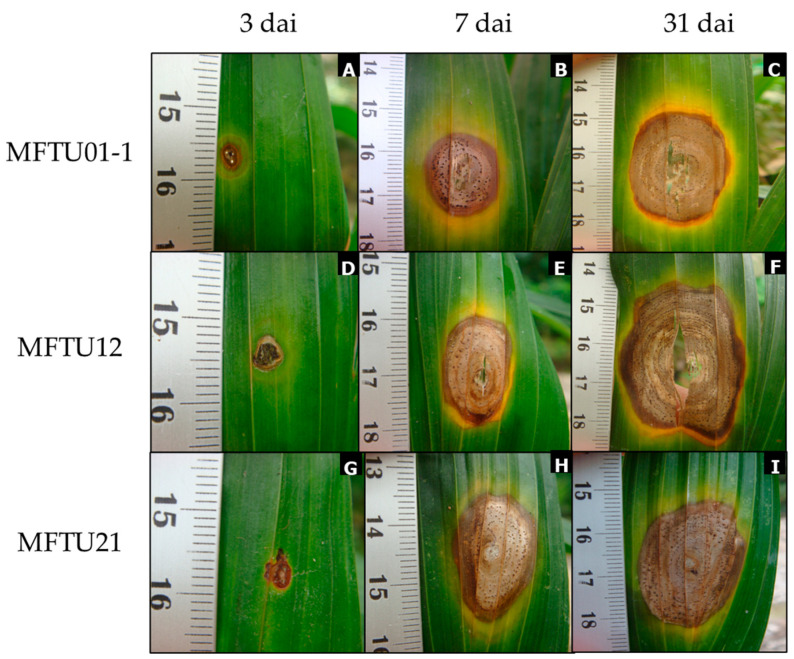
Pathogenicity test of *Pestalotiopsis* isolates. Leaf spot after inoculation, from left to right, 3, 17, and 31 days after inoculation. Isolates: MFTU01-1 (**A**–**C**); MFTU12 (**D**–**F**); and MFTU21 (**G**–**I**).

**Figure 6 jof-10-00024-f006:**
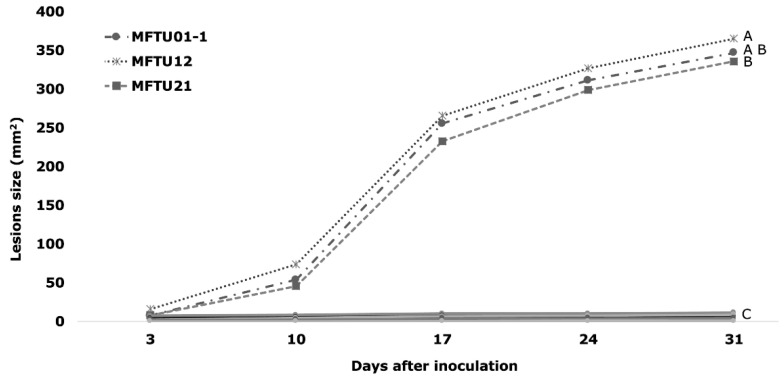
Development of symptoms in plants of the genotype Cereté × Deli after inoculation with *Pestalotiopsis* isolates. Different letters show significant differences according to the Tukey test (*p* < 0.05).

**Figure 7 jof-10-00024-f007:**
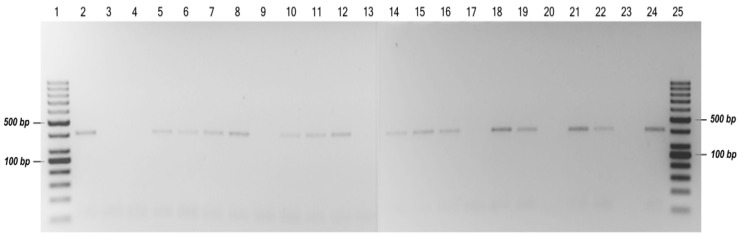
Agarose gel electrophoresis of PCR-amplified products using the specific primers Parenga155ITSF and Parenga534ITSR. Lanes 1 and 25–100 bp DNA ladder marker; lanes 2–22—amplified results with DNA samples from affected foliar tissues from the field; lane 23—negative control (healthy foliar tissue); and lane 24—positive control (isolate MFTU01-1).

**Table 1 jof-10-00024-t001:** Information on *Pestalotiopsis*, *Neopestalotiopsis*, and *Pseudopestalotiopsis* isolates obtained from the leaf blight of the oil palm hybrid (O × G).

Isolate Code	Genus	Host ^A^	Source	Geographic Origin
MFTU01-1	*Pestalotiopsis* sp.	Sinú (cerete) × Deli Yangambí	Leaf	Tumaco, Nariño, Colombia
MFTU04-3	*Neopestalotiopsis* sp.	Brazil × 7 polen africans	Leaf	Tumaco, Nariño, Colombia
MFTU06-1	*Neopestalotiopsis* sp.	Brazil × 7 polen africans	Leaf	Tumaco, Nariño, Colombia
MFTU06-2	*Neopestalotiopsis* sp.	Brazil × 7 polen africans	Leaf	Tumaco, Nariño, Colombia
MFTU06-3	*Neopestalotiopsis* sp.	Brazil × 7 polen africans	Leaf	Tumaco, Nariño, Colombia
MFTU07-1	*Neopestalotiopsis* sp.	Brazil × 7 polen africans	Leaf	Tumaco, Nariño, Colombia
MFTU07-2	*Neopestalotiopsis* sp.	Brazil × 7 polen africans	Leaf	Tumaco, Nariño, Colombia
MFTU12	*Pestalotiopsis* sp.	Sinú (cerete) × Deli Yangambí	Leaf	Tumaco, Nariño, Colombia
MFTU14-1	*Pseudopestalotiopsis* sp.	Coarí × La Mé	Leaf	Tumaco, Nariño, Colombia
MFTU14-2	*Pseudopestalotiopsis* sp.	Coarí × La Mé	Leaf	Tumaco, Nariño, Colombia
MFTU18	*Neopestalotiopsis* sp.	Manaos × Compacta	Leaf	Tumaco, Nariño, Colombia
MFTU21	*Pestalotiopsis* sp.	Coarí × La Mé	Leaf	Tumaco, Nariño, Colombia
MFTU25	*Neopestalotiopsis* sp.	Coarí × La Mé	Leaf	Tumaco, Nariño, Colombia
MFTU34-2	*Pseudopestalotiopsis* sp.	Brazil × 7 polen africans	Leaf	Tumaco, Nariño, Colombia
MFTU34-3	*Neopestalotiopsis* sp.	Brazil × 7 polen africans	Leaf	Tumaco, Nariño, Colombia
MFTU35	*Neopestalotiopsis* sp.	Coarí × La Mé	Leaf	Tumaco, Nariño, Colombia
MFTU36	*Neopestalotiopsis* sp.	Coarí × La Mé	Leaf	Tumaco, Nariño, Colombia
MFTU37	*Neopestalotiopsis* sp.	Coarí × La Mé	Leaf	Tumaco, Nariño, Colombia
MFTU38	*Neopestalotiopsis* sp.	Coarí × La Mé	Leaf	Tumaco, Nariño, Colombia
MFTU39-1	*Neopestalotiopsis* sp.	Coarí × La Mé	Leaf	Tumaco, Nariño, Colombia
MFTU39-2	*Neopestalotiopsis* sp.	Coarí × La Mé	Leaf	Tumaco, Nariño, Colombia
MFTU40-1	*Neopestalotiopsis* sp.	Coarí × La Mé	Leaf	Tumaco, Nariño, Colombia
MFTU41	*Neopestalotiopsis* sp.	Coarí × La Mé	Leaf	Tumaco, Nariño, Colombia
MFTU43	*Neopestalotiopsis* sp.	Coarí × La Mé	Leaf	Tumaco, Nariño, Colombia
MFTU44-2	*Neopestalotiopsis* sp.	Coarí × La Mé	Leaf	Tumaco, Nariño, Colombia

^A^ Oil palm genotype where infected foliar tissue was collected.

**Table 2 jof-10-00024-t002:** Cultural and morphological characteristics of 19 *Neopestalotiopsis*, 3 *Pestalotiopsis,* and 3 *Pseudopestalotiopsis* isolates obtained from the leaf blight of the oil palm hybrid.

Isolate Code	Genus/Species	Colony Type ^C^	Conidia Length (µm)		Conidia Width (µm)		Length of Basal Appendage (µm)		Median Cell Length (μm)		No. of Apical Appendages (Range)	Length of Apical Appendages (µm)		Color of Middle Cells
MFTU01-1	*Pestalotiopsis* sp.	3	28.1 ± 1.6 ^B^	a ^D^	6.6 ± 0.5	a	5.6 ± 1.3	b	18.2 ± 1.2	b	3–2	15.1 ± 2.6	c	Concolorous
MFTU04-3	*Neopestalotiopsis* sp.	5	24.4 ± 1.9	b	6.3 ± 0.6	b	5.4 ± 1.1	ab	16.3 ± 1.3	c	3–2	19.8 ± 3.5	b	Versicolorous
MFTU06-1	*Neopestalotiopsis* sp.	3	24.4 ± 1.1	b	6.5 ± 0.4	b	6.2 ± 0.9	ab	16.0 ± 1.3	c	3–2	23.3 ± 2.9	b	Versicolorous
MFTU06-2	*Neopestalotiopsis* sp.	2	28.8 ± 2.8	b	7.1 ± 0.5	b	5.1 ± 1.0	ab	18.7 ± 1.3	c	3–1	12.7 ± 2.7	b	Versicolorous
MFTU06-3	*Neopestalotiopsis* sp.	1	21.7 ± 1.8	b	5.9 ± 0.4	b	5.7 ± 1.1	ab	13.9 ± 1.2	c	3–2	17.0 ± 2.5	b	Versicolorous
MFTU07-1	*Neopestalotiopsis* sp.	2	22.0 ± 1.7	b	5.7 ± 0.4	b	4.6 ± 0.7	ab	14.8 ± 1.2	c	3–2	16.0 ± 2.2	b	Versicolorous
MFTU07-2	*Neopestalotiopsis* sp.	2	23.2 ± 1.5	b	5.7 ± 0.3	b	5.1 ± 0.9	ab	15.1 ± 1.1	c	3–2	18.0 ± 2.8	b	Versicolorous
MFTU12	*Pestalotiopsis* sp.	3	28.1 ± 2.3	a	6.5 ± 0.4	a	3.7 ± 0.6	b	17.8 ± 1.3	b	3–2	14.7 ± 2.4	c	Concolorous
MFTU14-1	*Pseudopestalotiopsis* sp.	4	26.1 ± 1.5	a	7.6 ± 0.5	a	4.2 ± 0.3	a	18.9 ± 1.0	a	4–2	19.2 ± 3.0	a	Concolorous
MFTU14-2	*Pseudopestalotiopsis* sp.	4	27.1 ± 1.4	a	6.7 ± 0.4	a	4.5 ± 0.5	a	18.8 ± 1.1	a	3–2	2.5 ± 3.8	a	Concolorous
MFTU18	*Neopestalotiopsis* sp.	5	20.9 ± 1.4	b	6.2 ± 0.3	b	4.9 ± 0.8	ab	13.7 ± 0.9	c	4–2	14.6 ± 2.5	b	Versicolorous
MFTU21	*Pestalotiopsis* sp.	3	28.6 ± 2.4	a	6.9 ± 0.4	a	4.7 ± 0.5	b	18.5 ± 1.2	b	3–2	15.7 ± 1.9	c	Concolorous
MFTU25	*Neopestalotiopsis* sp.	1	21.1 ± 1.3	b	6.4 ± 0.4	b	3.7 ± 0.5	ab	14.1 ± 0.9	c	3–2	20.1 ± 4.5	b	Versicolorous
MFTU34-2	*Pseudopestalotiopsis* sp.	4	31.7 ± 3.2	a	5.6 ± 0.5	a	6.6 ± 1.3	a	19.9 ± 1.9	a	3–2	22.9 ± 3.6	a	Concolorous
MFTU34-3	*Neopestalotiopsis* sp.	2	21.8 ± 1.7	b	5.8 ± 0.5	b	5.5 ± 1.0	ab	14.0 ± 0.9	c	3–2	21.7 ± 2.9	b	Versicolorous
MFTU35	*Neopestalotiopsis* sp.	1	22.9 ± 1.2	b	6.7 ± 0.5	b	5.1 ± 0.7	ab	14.9 ± 0.8	c	3–2	22.2 ± 5.0	b	Versicolorous
MFTU36	*Neopestalotiopsis* sp.	1	21.4 ± 1.2	b	6.5 ± 0.4	b	4.9 ± 0.6	ab	14.2 ± 0.8	c	3–2	21.3 ± 2.3	b	Versicolorous
MFTU37	*Neopestalotiopsis* sp.	1	19.8 ± 1.8	b	6.3 ± 0.4	b	5.4 ± 0.8	ab	13.3 ± 1.3	c	4–2	15.3 ± 2.8	b	Versicolorous
MFTU38	*Neopestalotiopsis* sp.	1	20.6 ± 1.1	b	5.9 ± 0.5	b	5.6 ± 0.9	ab	13.6 ± 0.9	c	4–2	21.3 ± 3.6	b	Versicolorous
MFTU39-1	*Neopestalotiopsis* sp.	1	21.9 ± 2.2	b	5.9 ± 0.5	b	5.7 ± 1.1	ab	14.5 ± 1.3	c	3–2	18.1 ± 2.9	b	Versicolorous
MFTU39-2	*Neopestalotiopsis* sp.	1	21.7 ± 1.7	b	5.3 ± 0.4	b	5.6 ± 0.7	ab	13.8 ± 1.2	c	3–2	15.9 ± 1.8	b	Versicolorous
MFTU40-1	*Neopestalotiopsis* sp.	6	27.8 ± 1.8	b	7.2 ± 0.7	b	3.5 ± 0.4	ab	19.1 ± 1.3	c	3–2	12.7 ± 1.9	b	Versicolorous
MFTU41	*Neopestalotiopsis* sp.	2	23.5 ± 1.9	b	7.1 ± 0.5	b	3.4 ± 0.5	ab	15.9 ± 1.1	c	3–1	13.7 ± 1.8	b	Versicolorous
MFTU43	*Neopestalotiopsis* sp.	1	21.5 ± 1.3	b	6.1 ± 0.3	b	4.9 ± 0.6	ab	14.1 ± 0.8	c	3–2	19.8 ± 2.7	b	Versicolorous
MFTU44-2	*Neopestalotiopsis* sp.	1	21.9 ± 1.8	b	6.1 ± 0.4	b	5.9 ± 0.9	ab	14.4 ± 1.0	c	3–2	19.5 ± 2.9	b	Versicolorous

a–c: The averages with the same letter are not significantly different. ^B^ Each value is the mean ± SE from measurements of 30 conidia. ^C^ Six colony morphotypes were as follows: 1 = cotton texture, irregular edges, high and limited appearance, white-grey (above) and white-salmon (reverse) colony, and development of surface conidiomata; 2 = cotton texture, irregular edges, high and limited appearance, white-cream (above) and white-salmon (reverse) colony, and development of surface conidiomata; 3 = cotton texture, irregular edges, flat and extended appearance, white-cream (above) and white-cream–red-orange (reverse) colony, and development of surface conidiomata; 4 = cotton texture, irregular edges, high and limited appearance, white-grey (above) and white-grey (reverse) colony, and development of surface conidiomata; 5 = cotton texture, irregular edges, flat and extended appearance, white-cream (above) and white-salmon (reverse) colony, and development of surface conidiomata; 6 = cotton texture, irregular edges, high and limited appearance, white-grey (above) and brown-reddish (reverse) colony, and development of surface conidiomata. ^D^ The average difference between each group is significant at 0.05 level; values in the same column with the same letters do not differ significantly according to Tukey’s test.

**Table 3 jof-10-00024-t003:** Details of isolates representing species in the phylogenetic clades of *Pestalotiopsis*, *Neopestalotiopsis*, and *Pseudopestalotiopsis* obtained in this study.

Isolate Code	Genus/Species	Host	Geographic Origin	GenBank Accession Number		
				*ITS*	*BTUB 2*	*TEF1-α*
MFTU01-1	*Pestalotiopsis arengae*	Sinú (cerete) × Deli Yangambí	Tumaco-Nariño, Colombia	MT952577	MT957907	MT957932
MFTU04-3	*Neopestalotiopsis* sp.	Brazil × 7 polen africans	Tumaco-Nariño, Colombia	MT952578	MT957908	MT957933
MFTU06-1	*Neopestalotiopsis* sp.	Brazil × 7 polen africans	Tumaco-Nariño, Colombia	MT952579	MT957909	MT957934
MFTU06-2	*Neopestalotiopsis* sp.	Brazil × 7 polen africans	Tumaco-Nariño, Colombia	MT952580	MT957910	MT957935
MFTU06-3	*N. surinamensis*	Brazil × 7 polen africans	Tumaco-Nariño, Colombia	MT952581	MT957911	MT957936
MFTU07-1	*Neopestalotiopsis* sp.	Brazil × 7 polen africans	Tumaco-Nariño, Colombia	MT952582	MT957912	MT957937
MFTU07-2	*Neopestalotiopsis* sp.	Brazil × 7 polen africans	Tumaco-Nariño, Colombia	MT952583	MT957913	MT957938
MFTU12	*P. arengae*	Sinú (cerete) × Deli Yangambí	Tumaco-Nariño, Colombia	MT952584	MT957914	MT957939
MFTU14-1	*Pseudopestalotiopsis cocos*	Coarí × La Mé	Tumaco-Nariño, Colombia	MT952585	MT957915	MT957940
MFTU14-2	*Ps. Cocos*	Coarí × La Mé	Tumaco-Nariño, Colombia	MT952586	MT957916	MT957941
MFTU18	*Neopestalotiopsis* sp.	Manaos × Compacta	Tumaco-Nariño, Colombia	MT952587	MT957917	MT957942
MFTU21	*P. arengae*	Coarí × La Mé	Tumaco-Nariño, Colombia	MT952588	MT957918	MT957943
MFTU25	*Neopestalotiopsis* sp.	Coarí × La Mé	Tumaco-Nariño, Colombia	MT952589	MT957919	MT957944
MFTU34-2	*Ps. ampullacea*	Brazil × 7 polen africans	Tumaco-Nariño, Colombia	MT952590	MT957920	MT957945
MFTU34-3	*Neopestalotiopsis* sp.	Brazil × 7 polen africans	Tumaco-Nariño, Colombia	MT952591	MT957921	MT957946
MFTU35	*Neopestalotiopsis* sp.	Coarí × La Mé	Tumaco-Nariño, Colombia	MT952592	MT957922	MT957947
MFTU36	*Neopestalotiopsis* sp.	Coarí × La Mé	Tumaco-Nariño, Colombia	MT952593	MT957923	MT957948
MFTU37	*Neopestalotiopsis* sp.	Coarí × La Mé	Tumaco-Nariño, Colombia	MT952594	MT957924	MT957949
MFTU38	*Neopestalotiopsis* sp.	Coarí × La Mé	Tumaco-Nariño, Colombia	MT952595	MT957925	MT957950
MFTU39-1	*Neopestalotiopsis* sp.	Coarí × La Mé	Tumaco-Nariño, Colombia	MT952596	MT957926	MT957951
MFTU39-2	*Neopestalotiopsis* sp.	Coarí × La Mé	Tumaco-Nariño, Colombia	MT952597	MT957927	MT957952
MFTU40-1	*Neopestalotiopsis* sp.	Coarí × La Mé	Tumaco-Nariño, Colombia	MT952598	MT957928	MT957953
MFTU41	*Neopestalotiopsis* sp.	Coarí × La Mé	Tumaco-Nariño, Colombia	MT952599	MT957929	MT957954
MFTU43	*Neopestalotiopsis* sp.	Coarí × La Mé	Tumaco-Nariño, Colombia	MT952600	MT957930	MT957955
MFTU44-2	*Neopestalotiopsis* sp.	Coarí × La Mé	Tumaco-Nariño, Colombia	MT952601	MT957931	MT957956

## Data Availability

Data are contained within the article and Appendix A.

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
