# Peer review of "Foliar Lesions Induced by Pestalotiopsis arengae in Oil Palm (O × G) in the Colombian Southwest Palm Zone"

_jof, 2023, doi:10.3390/jof10010024_

Round 1

Reviewer 1 Report

Comments and Suggestions for Authors

This manuscript describes the isolation and identification of the fungus Pestalotiopsis arengae as a pathogen that causes leaf blight on the oil palm in Colombia, by morphological characterization, pathogenicity test, and molecular and phylogenetic analysis. The findings are of significance that may guide the development of potential strategies for control of the disease. However, the manuscript falls short in providing solid and balanced evidence that P. arengae but not the Neopestalotiopsis sp. and Pseudopestalotiopsis sp., is the only pathogen responsible for leaf blight of the OxG hybrid oil palm in Columbia. Specifically, the field survey suggests that P. arengae is not the major pathogen for the leaf blight in the oil palm (3 samples out of 25 samples that were infected with Pestalotiopsis-like fungi), but pathogenicity assays showed that Neopestalotiopsis sp. and Pseudopestalotiopsis sp. were non-pathogenic on Sinú (cerete) x Deli Yangambí. The major issue remains: what are the agents responsible for blight symptoms of the 22 samples from which only N. sp. and P. sp. were isolated?

To improve the manuscript, issues were listed below:

Line 19: based on, not “on based”

Lines 19-20: “22 Pestalotiopsis-like isolates non-pathogenic”, these isolates should be assayed again on their native hybrids listed in Table 1 and Table 2

Line 36: 36,000, not 36.000 hectares

Lines 56-57: citation of reference: use family names only

Line 158: error in grammar for the sentence. Also, use “Sinú (cerete) x Deli Yangambí” instead of “Cerete x Deli”.

Line 164: error in grammar for the sentence.

Lines169-170: add temperature

Lines 184-185: why use initiators? Change to “primers”

Line 255: it would be better to write it as E. oleifera x E. guineensis (OxG)

Line 269: “Host A” - what does it mean?

Table 1: What is the relationship between the Hosts and OxG? Are all the hybrids from OxG? If so please state clearly in the text.

Table 1: only 3 out of 25 samples were Pestalotiopsis, a very low incidence rate. Should keep this rate in mind for data analysis and discussion.

Table 1: Neopestalotiopsis sp. and Neopestalotiopsis sp. were not isolated from Sinú (cerete) x Deli Yangambí. Why only Sinú (cerete) x Deli Yangambí was used in pathogenicity assay?

Table 2: it is a troublesome. Columns “Conidia width”, “Median cell length” and “Length of apical appendages” do not contain any real data; “Colour of middle cells: what do the figures mean?

Upper case A: Oil palm genotypes were not presented in the table

Significance test (Upper case D): should be marked right after the values which were not presented.

Line 314: Neopestalotiopsis (19 isolates), Pestalotiopsis (3 isolates); Pseudopestalotiopsis (3 isolates). Table 2 does not show Pestalotiopsis (6 isolates)

Line 318-319: “Subsequently, molecular analysis enabled the separation of a third genus called Pseudopestalotiopsis (3 isolates)” —- this sentence should be deleted, because one cannot state a conclusion before presentation of evidence.

Lines 326-345: data stated could not be found in Table 2.

Figure 3: no Scale bars on the images.

Line 361-369: should also use hybrid Coarí x La Mé in the pathogenicity assays, because Neopestalotiopsis and Pseudopestalotiopsis isolated were originally isolated from this variety.

Lines 376-377: “The other isolates induced small lesions but did not show development over time.” --- These results did not match those observed in the field (Figure 1 and Table 1). Was it due to different palm variety used?

Figure 4: Please mark clearly the isolates and times on the images. It seems that columns A, B, C are the days pdi (horizontal) and isolates should be marked vertically.

Lines 393-393: This sentence is not clear. Were the same microorganisms recovered from the lesions?

Figure 5: the lesion sizes were different from those in the text (lines 390) and Figure 6.

Figure 6 should be presented before Figure 5

Lines 408-418: This sentence is not correct in grammar and too long to read easily.

Lines 458-466: 15 of the 21 samples collected were P. arengae-positive (Figure 8). These data are in conflict with those in Table 2 where P. arengae were isolated from only 3 leaf samples. Were the samples the same as those in Table 1 or Table 2?

Discussion: Before the conclusion that Neopestalotiopsis and Pseudopestalotiopsis isolates were non-pathogenic, the authors should test the pathogenicity of these isolates on the host plants that they were originally sampled. The possible reasons that P. arengae was detected in much higher rate (15 out of 21 samples, Figure 8) than isolation rate (3 out of 25 samples, Table 2) should be carefully discussed.

Comments on the Quality of English Language

There are erroes in some paragraphs/sentences. The text needs to be carefully edited.

Author Response

Barrancabermeja-Colombia, November 28 of 2023

Dr. David S. Perlin

Editor-in-Chief, Journal of Fungi

The authors thank the reviewers for positive and constructive comments. A list of changes is provided on the attached revised version of the manuscript. Following the editor's recommendations, we have proofread the article using MDPI English Editing services (see attachment). Our detailed responses to the review’s comments are as follows:

Reviewer 1:

To improve the manuscript, issues were listed below:

Line 19: based on, not “on based”

>>> the word was corrected.

Lines 19-20: “22 Pestalotiopsis-like isolates non-pathogenic”, these isolates should be assayed again on their native hybrids listed in Table 1 and Table 2

>>> The pathogenicity of all isolates obtained was evaluated in all sampled cultivars, demonstrating that only Pestalotiopsis arengae is capable of developing foliar lesions. This confirms P. arengae as the causal agent of Pestalotiopsis in Colombia. These evaluations will be included in the supplementary information.

Line 36: 36,000, not 36.000 hectares

>>> the dot was changed.

Lines 56-57: citation of reference: use family names only

>>> the citation was corrected.

Line 158: error in grammar for the sentence. Also, use “Sinú (cerete) x Deli Yangambí” instead of “Cerete x Deli”.

>>> The grammatical error was corrected.

Line 164: error in grammar for the sentence.

>>> We adjusted the wording and fixed the grammatical error.

Lines169-170: add temperature.

>>> Temperature information was added.

Lines 184-185: why use initiators? Change to “primers”

>>> the word was changed.

Line 255: it would be better to write it as E. oleifera x E. guineensis (OxG).

>>> the word was changed.

Line 269: “Host A” - what does it mean?

>>> HOST A refers to the oil palm genotype from which infected foliar tissue was collected.

Table 1: What is the relationship between the Hosts and OxG? Are all the hybrids from OxG? If so please state clearly in the text.

>>> All hybrids are OxG, the word “OxG” was added in table 1.

Table 1: only 3 out of 25 samples were Pestalotiopsis, a very low incidence rate. Should keep this rate in mind for data analysis and discussion.

>>> Information was expanded in the discussion section (lines 548-553):

“Due to the low number of pure isolates, it was necessary to implement other methodologies, such as PCR, to detect the presence or absence of this microorganism. The low isolation rate is attributed to the presence of other secondary and opportunistic microorganisms in the affected tissues. These microorganisms were able to grow in the culture media used in our study because it was not selective for the isolation of P. arengae.”

Table 1: Neopestalotiopsis sp. and Neopestalotiopsis sp. were not isolated from Sinú (cerete) x Deli Yangambí. Why only Sinú (cerete) x Deli Yangambí was used in pathogenicity assay?

>>> Seeds of the Sinú (Cerete) x Deli Yangambí cultivar were used because they were the available material at the time of the test. However, additional evaluations were conducted, including these cultivars, and it was found that they are not pathogenic. As mentioned above, Pseudopestalotiopsis sp. and Neopestalotiopsis sp. were unable to develop lesions on any of the cultivars evaluated.

Table 2: it is a troublesome. Columns “Conidia width”, “Median cell length” and “Length of apical appendages” do not contain any real data; “Colour of middle cells: what do the figures mean?

Upper case A: Oil palm genotypes were not presented in the table

Significance test (Upper case D): should be marked right after the values which were not presented.

>>> The table was updated due to a format error, which resulted in the displacement of values and data. This error has been corrected, and the table now presents organized values and data.

Line 314: Neopestalotiopsis (19 isolates), Pestalotiopsis (3 isolates); Pseudopestalotiopsis (3 isolates). Table 2 does not show Pestalotiopsis (6 isolates).

>>> Initially, the isolates were classified based on their morphology into Neopestalotiopsis (19 isolates) and Pestalotiopsis (6 isolates). Subsequently, molecular biology tools were employed, revealing that 3 of the 6 isolates initially classified under Pestalotiopsis were correctly identified as Pseudopestalotiopsis. This clarification has been added to enhance the text's precision:

“Based on the initial morphological characterization, the isolates belonged to two genera: Neopestalotiopsis (19 isolates) and Pestalotiopsis (six isolates); this classification was assigned by the number of cells, the number of septa, and the color exhibited by the median cells of the conidia. Additionally, the length and width of the conidia, apical and basal appendages length, median cell length, and the number of basal and apical ap-pendages were determined and available in Table 2. Furthermore, molecular analysis (see Section 3.6) enabled the identification of a third genus, known as Pseudopestalotiopsis (comprising three isolates that were initially classified under the genus Pestalotiopsis based solely on their morphological characteristics)”

Line 318-319: “Subsequently, molecular analysis enabled the separation of a third genus called Pseudopestalotiopsis (3 isolates)” —- this sentence should be deleted, because one cannot state a conclusion before presentation of evidence.

>>> The paragraph was modified to clarify the reason why we included this information here.

Lines 326-345: data stated could not be found in Table 2.

>>> The table was updated due to a format error, which resulted in the displacement of values and data. This error has been corrected, and the table now presents organized values and data.

Figure 3: no Scale bars on the images.

>>> The image was corrected and the scale bar was added.

Line 361-369: should also use hybrid Coarí x La Mé in the pathogenicity assays, because Neopestalotiopsis and Pseudopestalotiopsis isolated were originally isolated from this variety.

>>> >>> The evaluation of the three previously missing cultivars (Brazil x 7 polen africans, Coarí x La Mé and Manaos x Compacta) has been added, and the corresponding information is available in the supplementary section. The results revealed that Neopestalotiopsis and Pseudopestalotiopsis isolates failed to exhibit pathogenicity when inoculated on the same cultivar from which they were isolated. This underscores a significant finding in this disease: only P. arengae isolates demonstrated pathogenicity across all evaluated cultivars, implicating this microorganism as the causal agent.

Lines 376-377: “The other isolates induced small lesions but did not show development over time.” --- These results did not match those observed in the field (Figure 1 and Table 1). Was it due to different palm variety used?

>>> In the evaluation with the Cereté x Delí cultivar, only one isolate, identified within the Neopestalotiopsis group (MFTU04-3), generated a small lesion; however, it did not progress over time. The wording was adjusted for clarity. Subsequent inoculations with other cultivars revealed no variation in the effect based on palm variety. The isolates that were pathogenic in the initial cultivar (Sinú Cerete x Delí) demonstrated pathogenicity for Brazil x 7 polen africans, Coarí x La Mé and Manaos x Compacta.

Figure 4: Please mark clearly the isolates and times on the images. It seems that columns A, B, C are the days pdi (horizontal) and isolates should be marked vertically.

>>> Modifications were made in Figure 4 to provide more clarity.

Lines 393-393: This sentence is not clear. Were the same microorganisms recovered from the lesions?

>>> Reisolation of the microorganism was obtained from the lesions, fulfilling Koch's postulates (line 419-422).

Figure 5: the lesion sizes were different from those in the text (lines 390) and Figure 6.

>>> The values that are written correspond to those represented in Figure 5 and refer to the average size of the lesion on day 31 dpi. The values in Figure 6 (bar graphs) represent the overall average lesion size from the first measurement to the last.

Figure 6 should be presented before Figure 5

>>> The order in which the figures are presented has been changed.

Lines 408-418: This sentence is not correct in grammar and too long to read easily.

>>> The wording was corrected to make it more fluid.

Lines 458-466: 15 of the 21 samples collected were P. arengae-positive (Figure 8). These data are in conflict with those in Table 2 where P. arengae were isolated from only 3 leaf samples. Were the samples the same as those in Table 1 or Table 2?

>>> Due to the difficulty of isolating P. arengae, additional work was carried out to design specific primers for the pathogen to enable molecular detection in the tissue, even when it was not isolated. This was confirmed through molecular detection of P. arengae in 15 of the 21 samples evaluated. Additionally, we clarify that the samples for this evaluation were again taken from plantations where the microorganisms evaluated were isolated:

“For this evaluation, at least one sample from each of the 16 plantations visited was used, and 21 samples were processed for the molecular detection of the pathogen.”

Discussion: Before the conclusion that Neopestalotiopsis and Pseudopestalotiopsis isolates were non-pathogenic, the authors should test the pathogenicity of these isolates on the host plants that they were originally sampled. The possible reasons that P. arengae was detected in much higher rate (15 out of 21 samples, Figure 8) than isolation rate (3 out of 25 samples, Table 2) should be carefully discussed.

>>> The isolates of Neopestalotiopsis and Pseudopestalotiopsis, which showed no pathogenicity on the Sinú (cerete) x Deli cultivar, were recently assessed on the other three cultivars (Brazil x 7 polen africans, Coarí x La Mé, and Manaos x Compacta), including the ones where they were initially isolated. This test showed that these isolates were non-pathogenic. This additional information has been included and can be found in the supplementary materials. A discussion of the potential reasons why P. arengae was detected at a much higher rate (15 out of 21 samples) using molecular biology tools has been added and expanded in the corresponding section. Additionally, it is important to clarify that molecular detection enabled us to identify the presence of the target microorganism; however, using non-selective enriched culture media for isolation permitted the growth of a wide variety of microorganisms.

Reviewer 2 Report

Comments and Suggestions for Authors

Dear authors, please find my comments to improve the manuscript below. 

Please remove citation (1) from the abstract 

The first paragraph (Line 27-30) of the introduction section is too short. It is better to combine it with the next paragraph. 

Paragraph 5 (Line 64-74) contains several similar monotonous sentences. Please re-write the paragraph and summarize the sentences. 

For clarity, it is better to separate Lines 76-78 into two sentences. 

The word 'phytosanitary' is not correctly used in line 94 since it refers to the measures and practices aimed at preventing the introduction, spread, and impact of plant pests and diseases. Please re-phrase. 

Line 110 - describe the criteria for initial observations. How old were the oil palms sampled in the study location? 

Line 113: Does the statement refer to artificially inoculated or natural infection? For the latter, how do the authors know exactly how many days after inoculation? Please justify and make amendments if relevant. 

Provide the reference/link for the Cellsens software (Line 149)

It is recommended that the authors re-write the first paragraph of the method section. The sentences are hard to follow. Also, please include more information on the cultivar used. How old were the plants? From where did the authors obtain the plantlets? Does the 'Preliminary test..' in Line 160 refer to the tests conducted by other researchers or by the authors but published elsewhere? The statement is confusing.  

Line 165 - What kind of tissue? Please specify

Line 179 - Pestalotiopsis-like?

2.6 Pestalotiopsis 'sp.' 

Suggestion for 3.1: Description of symptoms in naturally-infected farms 

Please include the pathogenicity test results for the other types of fungi since they were described in the result section. They can be included as supplementary files. 

Consider replacing Figure 7 with a sharper image. 

The authors are recommended to revise the article's title since it is not entirely about foliar lesions by P. arengae but more on identifying the causal agent for the specific disease symptoms of the oil palm hybrid as mentioned in the abstract.

Comments on the Quality of English Language

There are several mistakes in the formatting, sentence structures and grammar. Please proofread the article. 

Author Response

Barrancabermeja-Colombia, November 28 of 2023

Dr. David S. Perlin

Editor-in-Chief, Journal of Fungi

The authors thank the reviewers for positive and constructive comments. A list of changes is provided on the attached revised version of the manuscript. Following the editor's recommendations, we have proofread the article using MDPI English Editing services (see attachment). Our detailed responses to the review’s comments are as follows:

Reviewer 2:

Dear authors, please find my comments to improve the manuscript below. 

Please remove citation (1) from the abstract 

>>> The citation was removed from the abstract.

The first paragraph (Line 27-30) of the introduction section is too short. It is better to combine it with the next paragraph. 

>>> The two paragraphs were combined according to the reviewer's suggestions.

Paragraph 5 (Line 64-74) contains several similar monotonous sentences. Please re-write the paragraph and summarize the sentences. 

>>> Paragraph 5 was written and summarized according to the reviewer's suggestions.

For clarity, it is better to separate Lines 76-78 into two sentences. 

>>> Lines 76-78 were separated into two sentences for clarity.

The word 'phytosanitary' is not correctly used in line 94 since it refers to the measures and practices aimed at preventing the introduction, spread, and impact of plant pests and diseases. Please re-phrase. 

>>> The word 'phytosanitary' was changed and the wording of the sentence was corrected.

Line 110 - describe the criteria for initial observations. How old were the oil palms sampled in the study location? 

>>> The oil palms sampled at the study site were planted between 2007 and 2010. At the time of sampling, the palms were 6 to 9 years old. This information has been consolidated in this section.

Line 113: Does the statement refer to artificially inoculated or natural infection? For the latter, how do the authors know exactly how many days after inoculation? Please justify and make amendments if relevant. 

>>> We made the modification, the information on inoculation times was eliminated.

Provide the reference/link for the Cellsens software (Line 149).

>>> Reference was added.

It is recommended that the authors re-write the first paragraph of the method section. The sentences are hard to follow. Also, please include more information on the cultivar used. How old were the plants? From where did the authors obtain the plantlets? Does the 'Preliminary test..' in Line 160 refer to the tests conducted by other researchers or by the authors but published elsewhere? The statement is confusing.  

>>> The paragraph was adjusted, and the requested information was added. In addition, it was clarified that the preliminary inoculation trials were carried out by us.

Line 165 - What kind of tissue? Please specify

>>> The type of tissue was specified in the text and corresponds to leaves.

Line 179 - Pestalotiopsis-like?

>>> it was corrected based on the author's suggestion.

2.6 Pestalotiopsis 'sp.' 

>>> it was corrected based on the author's suggestion.

Suggestion for 3.1: Description of symptoms in naturally-infected farms 

>>> the author's suggestion was taken up and corrections were made to the manuscript.

Please include the pathogenicity test results for the other types of fungi since they were described in the result section. They can be included as supplementary files. 

>>> The results of pathogenicity tests for the other types of fungal isolates (Curvularia sp., Phoma sp. and Colletotrichum sp.) were added and are available in supplementary information.

Consider replacing Figure 7 with a sharper image. 

>>> The figure was replaced and improved to give more clarity and sharpness.

The authors are recommended to revise the article's title since it is not entirely about foliar lesions by P. arengae but more on identifying the causal agent for the specific disease symptoms of the oil palm hybrid as mentioned in the abstract.

>>>  We propouse this alternative title, according to the suggestion of the rewivier 2:

"Pestalotiopsis arengae, the causal agent of the leaf blight in interspecific hybrid cultivars OxG oil palm in the Colombian Southwestern zone"

Reviewer 3 Report

Comments and Suggestions for Authors

I believe the claims made by the authors in this manuscript.  However, the order of presentation of the Results mandates a very complex and confusing presentation.  The Results section needs to be revised. The molecular characterization data should be presented before the pathogenicity data, and that will enable a much more clear presentation of the pathogenicity data.  Both sections will have to be revised.

Some important details of method have been omitted and need to be added.  (See specific comments.)

Some sections of the text are very confusing and need to be simplified and clarified.  (See specific comments.)

L9-23.  The abstract reads very well and describes the results much more clearly than in the Results section.  The confusion in the Results section occurs because the pathogenicity data are presented before the molecular characterization data. 

L77.  It’s easier for the reader if the authors are identified in the citation. 

L92.  This foliar disease ?  (singular)

L110-116.  This paragraph is confusing and needs to be revised. I think the authors are attempting to put too much into one paragraph.  As result, I’m uncertain as to what was done. Is it true that symptoms on naturally infected leaves were recorded?  Is it also true that inoculations were done?  In English the adjective comes before the noun (not palms hybrid). 

L120.  What is an imprint?

L121-123.  How was the tray in a Ziploc bag made into a moist chamber?  Was water added?  Was there anything to distribute the water (i.e. paper towels or something similar)?

L129.  The “others” need to be identified here.

L131.  How did the authors determine that they transferred only a single spore to each Petri plate?

L147.  What is meant by the word “extracted”?  What is a conidial solution?  How was it made?  (Actually, “solution” is not the correct term; the conidia were “suspended” – presumably in water.) 

L158-162.  This paragraph needs to be revised.  Currently there are too many ambiguities.  Were there 25 isolates of each of the genera mentioned? The different inoculation techniques need to be described in words here.  The last sentence is a result. 

L164-168.  Was the 5 mm agar disc from the edge of the colony?  What type of wound was created?  Please describe the moist chamber – including the temperature.  Were whole plants inoculated? 

L179.  Do you mean Pestalotiopsis-like isolates? 

L217-220.  I’m uncertain as to what this sentence states.  The act of constructing primers cannot identify anything.  Please state explicitly what was done.

L260-264.  How many isolates of the other fungi were obtained? 

L273-287.  Are there photos of the six morphotypes?  Photos would help a lot.

L316-318.  Were these characteristics used to help separate the two genera?

L319. In which genus (genera) had the Pseudopestalotiopsis isolates been placed originally?

 L330-333.  This section needs to be revised.  As written, I do not know what it means.

L407-457.  This section should precede the section on pathogenicity.  This change would allow the pathogenicity data to be described much more clearly.  The overall result would be to clarify the presentation markedly.  The current order of presentation makes the pathogenicity section awkward and very difficult to understand.

L538 and following.  The numbers in parentheses are not clear to me.  What are they?  Are they min and max?  What does +/- SD mean? Is the standard deviation 8 or 0.4?  This notation needs to be clarified explicitly in the text.

Comments on the Quality of English Language

The English is largely okay, but I have suggested a few important changes in the specific comments.  

Author Response

Barrancabermeja-Colombia, November 28 of 2023

Dr. David S. Perlin

Editor-in-Chief, Journal of Fungi

The authors thank the reviewers for positive and constructive comments. A list of changes is provided on the attached revised version of the manuscript. Following the editor's recommendations, we have proofread the article using MDPI English Editing services (see attachment). Our detailed responses to the review’s comments are as follows:

Reviewer 3:

I believe the claims made by the authors in this manuscript.  However, the order of presentation of the Results mandates a very complex and confusing presentation.  The Results section needs to be revised. The molecular characterization data should be presented before the pathogenicity data, and that will enable a much more clear presentation of the pathogenicity data.  Both sections will have to be revised.

 Some important details of method have been omitted and need to be added.  (See specific comments.)

Some sections of the text are very confusing and need to be simplified and clarified.  (See specific comments.)

L9-23.  The abstract reads very well and describes the results much more clearly than in the Results section.  The confusion in the Results section occurs because the pathogenicity data are presented before the molecular characterization data. 

>>> The order of the methodology and results was organized in accordance with the comments.

L77.  It’s easier for the reader if the authors are identified in the citation.

   >>> The authors are identified in the citation.

L92.  This foliar disease ?  (singular)

 >>> The word was corrected.

L110-116.  This paragraph is confusing and needs to be revised. I think the authors are attempting to put too much into one paragraph.  As result, I’m uncertain as to what was done. Is it true that symptoms on naturally infected leaves were recorded?  Is it also true that inoculations were done?  In English the adjective comes before the noun (not palms hybrid). 

 >>> The paragraph was modified.

L120.  What is an imprint?

 >>> It refers to a direct verification of the structures of microorganisms present in the tissue, which are taken with a transparent tape, placed with lactophenol on a slide and then reviewed with the help of a microscope.

L121-123.  How was the tray in a Ziploc bag made into a moist chamber?  Was water added?  Was there anything to distribute the water (i.e. paper towels or something similar)?

 >>> The information the information was expanded to provide more clarity:

“To identify the fungal microorganisms present in the foliar tissue, imprints of the field-collected material were created. To induce sporulation, the tissue was placed in a moist chamber consisting of 28-oz aluminum trays. Each tray contained a paper towel moistened with sterile, distilled water and was covered with a piece of tulle to prevent direct contact between the leaves and the water. The trays were then placed inside Ziploc© bags.”

L129.  The “others” need to be identified here.

 >>> The culture media were added.

L131.  How did the authors determine that they transferred only a single spore to each Petri plate?

 >>> To make the monosporic cultures, serial dilutions of a spore suspension were made, 20 microliters were taken from the dilutions and placed on PDA medium with the help of a Digralsky loop. After four hours the plates were checked under the microscope to verify the germination of the conidia, once germination was observed with the help of a sterilized loop was transferred to another Petri dish with PDA medium to obtain the isolate.

We are based on research by Khadidja K. Isolation, characterization of phytopathogenic fungi Alternaria sp, and physico-chemical study. Faculty of Natural and Life Science Department of Biology. 2020.

L147.  What is meant by the word “extracted”?  What is a conidial solution?  How was it made?  (Actually, “solution” is not the correct term; the conidia were “suspended” – presumably in water.) 

 >>> The sentence was corrected.

L158-162.  This paragraph needs to be revised.  Currently there are too many ambiguities.  Were there 25 isolates of each of the genera mentioned? The different inoculation techniques need to be described in words here.  The last sentence is a result. 

 >>> The paragraph was revised and the information about isolates was added.

L164-168.  Was the 5 mm agar disc from the edge of the colony?  What type of wound was created?  Please describe the moist chamber – including the temperature.  Were whole plants inoculated? 

 >>> The information was added. About the conditions: The humid chamber was made by placing the plants inside a plastic bag and ensuring humidity with cotton sterilized and moistened with sterilized distilled water.

Whole plants were used.

L179.  Do you mean Pestalotiopsis-like isolates? 

>>> Yes, it is.

L217-220.  I’m uncertain as to what this sentence states.  The act of constructing primers cannot identify anything.  Please state explicitly what was done.

>>> To clarify, during primer design, we designed specific primers for the detection of the target microorganism, Pestalotiopsis arengae. The purpose of this process was to enable molecular detection of the pathogen in the plant tissue, even though it was not successfully isolated, due to high presence of secondary microorganism. For this reason, a comparison of the ITS region of P. arengae isolates and different sequences of Pestalotiopsis, Neopestalotiopsis, and Pseudopestalotiopsis was carreid out. The regions were taken for the synthesis of the primers, where it allowed a correct alignment only for the Pestalotiopsis species.

L260-264.  How many isolates of the other fungi were obtained? 

 >>> The information was added.

L273-287.  Are there photos of the six morphotypes?  Photos would help a lot.

>>> No, only we added photos the pathogenic isolates.

L316-318.  Were these characteristics used to help separate the two genera?

>>> Yes,  these morphological characteristics helped to separate the two genera.

L319. In which genus (genera) had the Pseudopestalotiopsis isolates been placed originally?

>>> This information was added in the lines 340 -348:

“Based on the initial morphological characterization, the isolates belonged to two genera: Neopestalotiopsis (19 isolates) and Pestalotiopsis (six isolates); this classification was assigned by the number of cells, the number of septa, and the color exhibited by the median cells of the conidia. Additionally, the length and width of the conidia, apical and basal appendages length, median cell length, and the number of basal and apical appendages were determined and available in Table 2. Furthermore, molecular analysis (see Section 3.5) enabled the identification of a third genus, known as Pseudopestalotiopsis (comprising three isolates that were initially classified under the genus Pestalotiopsis based solely on their morphological characteristics)”.

L330-333.  This section needs to be revised.  As written, I do not know what it means.

>>>> An adjustment was made to the paragraph, and it was specified what each of the measures corresponded to.

L407-457.  This section should precede the section on pathogenicity.  This change would allow the pathogenicity data to be described much more clearly.  The overall result would be to clarify the presentation markedly.  The current order of presentation makes the pathogenicity section awkward and very difficult to understand.

>>>> This section was reorganized according to your suggestions.

L538 and following.  The numbers in parentheses are not clear to me.  What are they?  Are they min and max?  What does +/- SD mean? Is the standard deviation 8 or 0.4?  This notation needs to be clarified explicitly in the text.

>>>> These values correspond to morphological measurements, based on a certain number of structures evaluated. The values correspond to the average of the minimum and maximum of these measurements. The numbers in parentheses correspond to the measurement furthest from the group, either from the maximum or the minimum.

>>>> The standard deviation SD is the number before the sign ±, after these, represents how it can fluctuate.

Comments on the Quality of English Language

The English is largely okay, but I have suggested a few important changes in the specific comments.  

>>>>We thank you for your comments, most of which have been added. Likewise, a style correction was made.

Reviewer 4 Report

Comments and Suggestions for Authors

The manuscript presented by Betancourt et al. and entitled Foliar lesions induced by Pestalotiopsis arengaein oil palm (OxG) in the Southwestern Colombian Zone, focuses on identifying and describing the symptomatology of bud rot diseases affecting the oil palm hybrid Elaeis oleífera x Elais guuineensis. The authors performed pathogenicity tests with the obtained microorganisms and characterized the causal agent of the disease.

Curvularia,Colletotrichum, Phoma, and 25 Pestalotiopsis-like fungi were isolated from the foliar lesions. In the pathogenicity tests, the symptoms observed in the field were reproduced by Pestalotiopsis arengae. Interestingly, this is the first report of P. arengaeas as the causal agent of foliar lesions in OxG hybrid oil palm in Colombia.

Considering that worldwide there is a lack of information on the causal agents of the disease, this manuscript presents novel and useful information that could contribute to mitigate the incidence and severity of the disease.

The results are interesting, but the methods section must be improved. Please consider including the following:

-Climatic characteristics of the study area. There is a mention of precipitation and annual temperature, but how is the annual seasonality? Is there a strong difference between the rainy and dry seasons? I am asking this, as I do not know if there could be differences in disease incidence depending on the season of the year.

On line 110 you mention that under field conditions, initial observations were made to select palms hybrid to be evaluated, but how did you select the palms? Focusing only on those with disease symptoms, or did you use particular sampling methods? In which year and season the study was performed?

Regarding the pathogenicity tests it would be useful to know how you obtained the plants to be tested. Did you get them from seed? How old were they when you performed the tests?

Author Response

Barrancabermeja-Colombia, November 28 of 2023

Dr. David S. Perlin

Editor-in-Chief, Journal of Fungi

The authors thank the reviewers for positive and constructive comments. A list of changes is provided on the attached revised version of the manuscript. Following the editor's recommendations, we have proofread the article using MDPI English Editing services (see attachment). Our detailed responses to the review’s comments are as follows:

Reviewer 4:

The manuscript presented by Betancourt et al. and entitled Foliar lesions induced by Pestalotiopsis arengae in oil palm (OxG) in the Southwestern Colombian Zone, focuses on identifying and describing the symptomatology of bud rot diseases affecting the oil palm hybrid Elaeis oleífera x Elais guuineensis. The authors performed pathogenicity tests with the obtained microorganisms and characterized the causal agent of the disease.

Curvularia,Colletotrichum, Phoma, and 25 Pestalotiopsis-like fungi were isolated from the foliar lesions. In the pathogenicity tests, the symptoms observed in the field were reproduced by Pestalotiopsis arengae. Interestingly, this is the first report of P. arengae as as the causal agent of foliar lesions in OxG hybrid oil palm in Colombia.

Considering that worldwide there is a lack of information on the causal agents of the disease, this manuscript presents novel and useful information that could contribute to mitigate the incidence and severity of the disease.

The results are interesting, but the methods section must be improved. Please consider including the following:

-Climatic characteristics of the study area. There is a mention of precipitation and annual temperature, but how is the annual seasonality? Is there a strong difference between the rainy and dry seasons? I am asking this, as I do not know if there could be differences in disease incidence depending on the season of the year.

>>> According to the Meteorological Station located in the El Mira Research Center of Agrosavia (observation window of the last 20 years), Tumaco has an average annual precipitation of 2,910 mm distributed mainly in a single wet season between the months of January to June. Likewise, the average annual temperature is 26.6°C with a maximum of 27.8°C and a minimum of 25.6°C. In the case of the disease, the summer season favors the life cycle of the insect that induces the disease.

On line 110 you mention that under field conditions, initial observations were made to select palms hybrid to be evaluated, but how did you select the palms? Focusing only on those with disease symptoms, or did you use particular sampling methods? In which year and season the study was performed?

>>> A visit was made to different plantations and a general check was made, to then focus on the palms that had some type of foliar lesion caused by fungi. Subsequently, isolations were made, where morphotypes similar to pestalotiopsis were found. From there, we went deeper and focused on obtaining more isolates from leaf spots of palms with the same symptomatology. This work was carried out during half of 2018 until 2020.

Regarding the pathogenicity tests it would be useful to know how you obtained the plants to be tested. Did you get them from seed? How old were they when you performed the tests?

>>> Seedlings were obtained from seeds of Sinú (cerete) x Deli Yangambí (from Agrosavia El Mira, Colombia). At the time of inoculation, the plants were approximately 5-7 months old.

Round 2

Reviewer 1 Report

Comments and Suggestions for Authors

1. Please check the legend to the Fig. 1, it does not seem to be correct.

2. Lesion sizes of isolates MFTU01-1, MFTU12, and MFTU21 (Pestalotiopsis arengae) were from 336 mm2 to 365 mm2 (Fig.6), however, lesion sizes of the same isolates were from about 170 mm2 to about 210 mm2. Why? Needs to explain the reasons in the text.

3. Supplementary Fig.8: lesion size (mm), changed to (mm2); Tucky’s test (a<0.05), changed to (p<0.05)

Author Response

Barrancabermeja-Colombia, December 12 of 2023

Dr. David S. Perlin

Editor-in-Chief, Journal of Fungi

The authors thank reviewer 1 for his valuable contributions in revising and improving the manuscript. Following the editor's recommendations, we have proofread the article using MDPI English Editing services (see attachment).

Our detailed responses to the review’s comments are as follows:

Reviewer 1:

  1. Please check the legend to the Fig. 1, it does not seem to be correct.

>>> the legend was changed.

  1. Lesion sizes of isolates MFTU01-1, MFTU12, and MFTU21 (Pestalotiopsis arengae) were from 336 mm2to 365 mm2(Fig.6), however, lesion sizes of the same isolates were from about 170 mm2 to about 210 mm2. Why? Needs to explain the reasons in the text.

>>> The figure 7 was deleted to no create confusion to the readers.

  1. Supplementary Fig.8: lesion size (mm), changed to (mm2); Tucky’s test (a<0.05), changed to (p<0.05)

>>> the supplementary figure 8 was modified.